# The impact of coupled 3D shortwave radiative transfer on surface radiation and cumulus clouds over land

Mirjam Tijhuis[1], Bart J. H. van Stratum[1], and Chiel C. van Heerwaarden[1]

[1]Meteorology and Air Quality Group, Wageningen University & Research, Wageningen, the Netherlands

**Correspondence:** Mirjam Tijhuis (mirjam.tijhuis@wur.nl)

**Abstract.** Radiative transfer is a 3D process, but most atmospheric models consider radiation only in the vertical direction for computational efficiency. This results in inaccurate surface radiation fields, as the horizontal transport of radiation is neglected. Previous work on 3D radiative effects mainly used 3D radiative transfer uncoupled from the flow solver. In contrast, our current work uses 3D radiative transfer coupled to the flow solver to study its impact on the development of clouds and the resulting impact on the domain-averaged surface solar irradiance. To this end, we performed a series of realistic Large-Eddy simulations with MicroHH. To improve the level of realism of our radiation, we first included the direct effect of aerosols using aerosol data from the CAMS global reanalysis. Next, we performed simulations with 1D radiative transfer and with a coupled ray tracer, for 12 days on which shallow cumulus clouds formed over Cabauw, the Netherlands. In general, simulations with the coupled ray tracer have a higher domain-averaged liquid water path, larger clouds, and similar cloud cover compared to simulations with 1D radiative transfer. Furthermore, the domain-averaged direct radiation is decreased with 3D radiative transfer and the diffuse radiation is increased. However, the average difference in global radiation is less than $1 \, \mathrm{W \, m^{-2}}$, as the increase in global radiation from uncoupled 3D radiative transfer is counterbalanced by a decrease in global radiation caused by changes in cloud properties.

## 1 Introduction

Radiative transfer is a 3D process, which is often reduced in atmospheric models to a 1D process for computational efficiency. Previous work has shown that this 1D approximation leads to errors as it neglects the horizontal transport of radiation (e.g., Várnai and Davies, 1999; Marshak and Davis, 2005; Cahalan et al., 2005; Pincus et al., 2005; Gristey et al., 2020a). These errors occur when the horizontal resolution is such that the cloud shadows are located in different grid cells than the clouds, which is the case in Large-Eddy Simulation (LES) models and can be the case in cloud resolving models. In these models, simulations with 1D radiation have brighter cloud shadows than simulations with 3D radiation, with cloud shadows that are always positioned directly underneath the clouds.

In addition, simulations with 1D radiation do not capture cloud enhancements, which are peaks in surface radiation that exceed the clear sky radiation. These differences in surface radiation are relevant for multiple processes, such as renewable energy production (e.g., Kreuwel et al., 2020) and photosynthesis (e.g., Kanniah et al., 2012; Vilà-Guerau de Arellano et al., 2023). Furthermore, the differences in radiation can have an impact on the development of clouds, mainly caused by differences in

radiation at the surface and the corresponding differences in surface heat fluxes (Veerman et al., 2022, 2020; Jakub and Mayer, 2017). This impact of 3D radiation on clouds can only be captured when the results of the radiative transfer calculations impact the surface and atmosphere, thus when 3D radiative transfer is coupled to the flow solver in simulations with an interactive land-surface. As it recently became possible to do Large-Eddy Simulations (LES) with coupled 3D radiation (Veerman et al.,
2022; Jakub and Mayer, 2015), we now have the opportunity to investigate the impact of coupled 3D radiative transfer on clouds and surface radiation. Therefore, this paper systematically compares the influence of coupled 1D and 3D radiation on surface radiation and clouds.

Previous studies with 3D radiative transfer that was not coupled to the flow solver (i.e. uncoupled 3D radiative transfer) showed that including the 3D radiative effect is essential to model the correct spatial distribution of shortwave radiation at
the surface, including cloud enhancements (Gristey et al., 2020a; Tijhuis et al., 2023). In addition, Gristey et al. (2020a) showed that uncoupled 3D radiative transfer changes the domain-averaged shortwave radiation compared to 1D radiation. The differences between 1D and uncoupled 3D radiative transfer can be explained by two opposing changes. On the one hand, the direct radiation is reduced as the cloud shadow area increases (side illumination, Hogan and Shonk (2013)). On the other hand, the diffuse radiation at the surface increases because radiation escapes from the sides of clouds (Hogan and Shonk, 2013;
Várnai and Davies, 1999) and radiation gets entrapped between cloud layers and between the clouds and the surface (Hogan et al., 2019). Uncoupled 3D radiative transfer has also previously been used for the validation of several approximations of 3D radiative transfer (Gristey et al., 2020b, 2022; Wissmeier et al., 2013; Wapler and Mayer, 2008; Jakub and Mayer, 2015; Hogan et al., 2016).

The differences in the domain-averaged surface shortwave radiation and its spatial distribution will impact the development
of the clouds, which can only be captured with coupled 3D radiation. So far, the impact of cloud shadows on the development of clouds has mainly been shown with idealized cases (e.g., Gronemeier et al., 2017; Lohou and Patton, 2014; Horn et al., 2015; Schumann et al., 2002). Horn et al. (2015) and Schumann et al. (2002) demonstrated that cloud shadows reduce cloud size and lifetime compared to a situation without cloud shadows, whereas Gronemeier et al. (2017) compared different solar zenith angles and found that the differences in cloud shadows cause smaller clouds at smaller solar zenith angles and larger clouds at
larger solar zenith angles. Lohou and Patton (2014) showed that the surface heterogeneities caused by cloud shadows influence the fluxes up to the height of the cloud roots. As Gronemeier et al. (2017) already pointed out, these studies were limited to fixed solar zenith and azimuth angles, fixed background wind speeds and wind directions. In addition, these studies all simplified radiation either by horizontally shifting the cloud shadow (Schumann et al., 2002) or by using 1D radiation (Lohou and Patton, 2014; Horn et al., 2015) or tilted columns (Gronemeier et al., 2017). Besides the 3D effects in the shortwave spectral range,
Klinger et al. (2017) found that 3D effects in the longwave spectral range can cause larger clouds when using an approximation for 3D longwave radiation. An example of how coupled 3D radiation can influence clouds is provided by Jakub and Mayer (2017), who demonstrated that coupled 3D radiative transfer impacts the formation of idealized cloud streets for a range of background winds, solar zenith and azimuth angles. These cloud streets form perpendicular to the prescribed solar incidence angle, and when the solar incidence angle is changed 90 degrees, the cloud streets change orientation accordingly in about one
hour. Veerman et al. (2020) and Veerman et al. (2022) studied realistic cases of shallow cumulus including large scale forcing

and the daily cycle of solar zenith and azimuth angle. They showed that coupled 3D radiation results in larger and thicker clouds, but these results are based on single case studies.

We aim to systematically investigate the impact of coupled 3D shortwave radiative transfer on the mean surface radiation and the development of clouds. To this end, we use an LES model (MicroHH, van Heerwaarden et al. (2017)) with interactive land surface and a coupled ray tracer for the shortwave radiation. We focus on cumulus clouds, as they cause large variability in surface radiation and are strongly coupled to the surface. First, we implemented aerosol optics in our radiative transfer solver to reduce a systematic bias in the surface radiation partitioning caused by the absence of aerosols in the radiation model. To validate the inclusion of aerosols, we performed simulations for a set of days with clear skies over Cabauw, the Netherlands, which we compared with observations (section 3). Next, we used the setup with aerosols to simulate a set of 12 days during which shallow cumulus clouds developed. After comparing the results with observations to ensure that the simulations resemble reality, we used these simulations to study the impact of coupled 3D radiative transfer on cloud properties and surface direct, diffuse, and global radiation. To understand the impacts of coupled 3D radiative transfer better, we also investigate the impact of uncoupled 3D radiative transfer, and we examine how the changes in cloud properties feed back to the surface radiation.

## 2 Methods

### 2.1 Case selection

For this study, we selected two sets of cases: one set with clear sky days to test our implementation of the aerosol optics and one set with cumulus days to study the impact of coupled 3D radiative transfer. To select the cases, we used the dataset of Mol et al. (2023), which provides ten years of solar irradiance observations in Cabauw, together with, among others, a classification of the weather (clear-sky, variable, overcast) and the satellite-derived cloud type (for the years 2014-2016). Using this dataset, we first selected 13 clear sky days. We chose to use clear sky days to validate our aerosol implementation, as it allows for a direct comparison between observations and our simulations, which is not possible for cloudy days because of the stochastic nature of the clouds. From the days with at least half of the day classified as clear sky, we manually selected 13 days which cover all wind directions and a range of aerosol optical depths (0.015-0.5) that covers all values that are generally found for Cabauw (van Heerwaarden et al. (2021), their figure 3c). Next, we selected 12 days with cumulus clouds. We selected the days in the period 2014-2016 with at least 5 hours classified as cumulus and no near-overcast conditions with a cloud cover larger than 95%, which results in a total of 20 days. These days were simulated with MicroHH (using 1D radiative transfer) to select the days where the simulated cloud cover visually matches the observed cloud cover, meaning that there is no systematic under or overestimation of the cloud cover by tens of percents, which can happen e.g. when the clouds are forced by a large-scale system that is not captured by the simulation. This resulted in a selection of 12 days, from which the last hours (17-21 UTC) of 4 July 2016 were excluded from further analysis, as the clouds are not surface driven during these hours.

## 2.2 Model simulations

### 2.2.1 General setup

We used MicroHH (van Heerwaarden et al., 2017) to perform realistic LESs. Here we only describe the model domain and the settings that regard the radiative transfer, as it is practically unfeasible to describe all model settings of these LESs. Our complete model setup can be found on Zenodo (https://doi.org/10.5281/zenodo.11234716).

For the clear sky days, we used a domain size of 12.8 x 12.8 x 4 km$^3$, with a horizontal resolution of 50 m and a vertical resolution of 25 m and the simulations run from 6 - 18 UTC. For each day, we performed two simulations: one with and one without aerosols. Our simulations of the cumulus days have the same horizontal and vertical resolution as the clear-sky days, but a larger domain size of 25.6 x 25.6 x 6.4 km$^3$ and they run longer, from from 3 - 21 UTC. In all simulations, we used an interactive land-surface scheme, similar to HTESSEL (Balsamo et al., 2009). This scheme calculates the surface fluxes, temperature and humidity using four soil layers with a skin layer on top. This skin layer represents the vegetation which intercepts the radiation and responds instantaneously (in other words it has a zero heat capacity), after which part of the heat is conducted to the underlying soil layers that respond slower. In addition, the land-surface scheme takes into account changes in the canopy resistance as a function of incoming shortwave radiation, soil moisture, and vapor pressure deficit. Regarding the vegetation and the soil we use the same settings as (van Stratum et al., 2023), where it is shown that shallow convection is realistically modelled with this setup. Initial and boundary conditions were derived from ERA5 using (LS)$^2$D (van Stratum et al., 2023).

In all simulations radiation is calculated every minute using RTE+RRTMGP (Pincus et al., 2019)(1D radiative transfer) and for the 3D radiation, we used the ray tracer of Veerman et al. (2022). This ray tracer uses the power of graphics processing unit computing to perform Monte Carlo ray tracing coupled to the flow solver. Calculating the radiation more often (every 15 seconds) has a limited impact on the results (not shown) and therefore the one minute time step was chosen to limit the computational costs. For the impact of gases on radiation, we used time and height dependent water vapor from our simulations, ozone from ERA5 (Hersbach et al., 2020), and carbon monoxide and methane from the CAMS global greenhouse gas reanalysis (Inness et al., 2019a). The other gases that influence radiation are assumed constant and were taken from RFMIP (Pincus et al., 2016). As our domain top is at 6.4 km, MicroHH accounts for the impact of gases and aerosols on radiation above our domain top, by calculating radiation for one column that extends until the top of the atmosphere. This background column consists of time and height dependent ERA5 data (temperature, pressure, water vapor, and ozone at the native ERA5 model levels), combined with time and height dependent carbon monoxide and methane from CAMS, and constant gases from RFMIP.

For each cumulus day, we performed two simulations: one with coupled 1D radiation and one with coupled 3D radiation. In these coupled simulations, the results of the radiative transfer calculations impact the surface and atmosphere. Here (and in the remainder of this work), *1D radiation* is short for 1D radiative transfer which means that we used the two-stream approach and *3D radiation* is short for 3D radiative transfer which means that we used the ray tracer. We performed 3 simulations of each cumulus day/coupled radiation method with a different random seed for the initial random perturbations, to estimate statistical convergence of our simulation results. Our plots show the average result of the 3 simulations, unless indicated otherwise.

### 2.2.2 Link between radiation and clouds

To understand the differences in clouds between the simulations with coupled 1D and 3D radiation, we investigated the link between radiation and clouds. Based on previous research (Veerman et al., 2020; Jakub and Mayer, 2017; Veerman et al., 2022), we hypothesized that changes in the distribution of surface radiation alter the surface energy balance, which modifies the updrafts that link the surface to the clouds, resulting in a change in clouds. These changes occur because with 3D radiation the cloud shadows are displaced, thus cloud shadows are not directly below the clouds and cloud enhancements can occur. In the shadows, the fluxes are reduced, and in the cloud enhancements the fluxes are increased. These differences in surface fluxes determine where updrafts are likely to form, and therefore where clouds grow.

Our hypothesis also shows the complexity of the problem. Changes in clouds will affect the surface radiation, which makes it complex to determine what is the cause and what is the consequence, and it is hard to prove any causality. However, we can investigate if our simulations support our hypothesis by looking at the correlations between the cloud shadow displacement and the changes in clouds. Since the changes in liquid water path, cloud cover and cloud depth are related, we only examined the relative difference in liquid water path, which is the difference in liquid water path between the simulations with 3D and 1D radiation relative to the liquid water path in the simulation with 1D radiation. We described the cloud shadow displacement relative to the cloud with three factors: 1. The distance between the cloud and its shadow (derived from the domain-averaged cloud base height and solar zenith angle); 2. The angle between the sun and the wind (derived from the wind direction at 500 m and the solar azimuth angle); 3. The wind speed at 500 m.

To test our hypothesis, we focused on the times between cloud onset and the time with the maximum domain-averaged liquid water path. After a similar cloud onset, the clouds in simulations with 1D and 3D radiation can start to differ as 3D radiative effects start to play a role, so we only investigated times after cloud onset. Later on, dissipation of the clouds starts to play an important role in the development of the cloud field. As we expect the clouds in the simulations with 3D radiation to be thicker and larger (Veerman et al., 2020, 2022), they will dissipate slower. We argue that this is mainly important after the maximum domain-averaged liquid water path is reached, and therefore we only investigated times before this maximum.

### 2.2.3 Uncoupled radiation computations

To understand the differences in radiation between the simulations with coupled 1D and 3D radiation, we performed additional uncoupled radiation computations, which means that the results of the radiative transfer calculations do not impact the surface and atmosphere. Our setup is shown schematically in Fig. 1, where the simulations with coupled 3D radiation and coupled 1D radiation as described in Sect. 2.2.1 are located in the top left and bottom right of the schematic. The impact of coupled 3D radiation, hereafter referred to as the coupled effect, is the difference between the simulations with coupled 1D and 3D radiation, which is indicated with the blue line in Fig.1 and labeled 3D-1D.

We calculated two types of uncoupled radiation: uncoupled 3D radiation and uncoupled 1D radiation. For the uncoupled 3D radiation, indicated in the bottom left of Fig. 1 ($1D_{rad3D}$), we took the cloud fields from the simulations with coupled 1D radiation and perform offline 3D radiation computations. This approach is similar to what has been done in previous studies

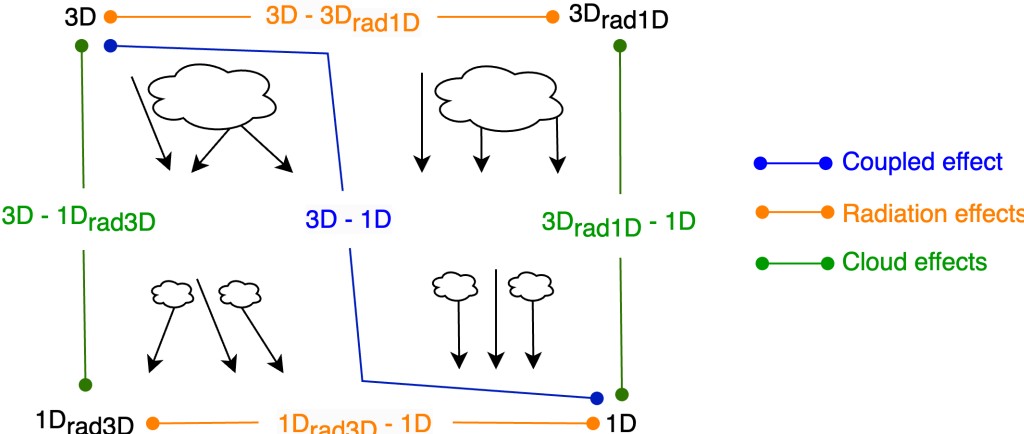

**Figure 1.** Schematic overview of the simulations with coupled and uncoupled radiation. The simulation with coupled 3D radiation is in the top left (3D), the simulation with coupled 1D radiation is in the bottom right (1D). The top right is the uncoupled 1D radiation of our simulation with coupled 3D radiation, the bottom left is the uncoupled 3D radiation of our simulation with coupled 1D radiation. The blue line indicates the coupled effect, the orange lines the two different radiation effects, and the green lines the two different cloud effects.

(e.g., Gristey et al., 2020a, b). For the uncoupled 1D radiation ($3D_{rad1D}$), indicated in the top right of Fig. 1, we took the cloud fields from the simulations with coupled 3D radiation and perform offline 1D radiation computations.

We used these additional calculations to split the difference between 3D and 1D in two parts. The first part is the radiation effect, which is indicated with the orange lines in Fig. 1. This is the difference in radiation that occurs when the radiative transfer method differs, but the clouds are the same. The same radiation effect using uncoupled 3D radiation was studied before by Gristey et al. (2020a).

     The second part is the cloud effect, which is indicated with the green lines in Fig. 1. This is the difference in radiation that
occurs when the clouds are different, but the radiative transfer method is the same. We are aware that physically the two effects can not be seen separate from each other, but purely from a mathematical point of view, the two effects add up to the coupled effect, and the separate effects are easier to understand, as we will show in Sect. 4.2. We obtained both the radiation effect and cloud effect in two ways, using either $3D_{rad1D}$ or using $1D_{rad3D}$, as is visible in Fig. 1. In Sect. 4.2, we will show how the radiation effect and cloud effect differ between the two ways of splitting.

### 2.2.4 Aerosols

Previous studies have shown the importance of aerosols when simulating radiation realistically (Schmidt et al., 2009; Gristey et al., 2022; Tijhuis et al., 2023). Therefore, we created the option in MicroHH to use aerosol data from the CAMS global reanalysis to include the direct effect of aerosols in our simulations, hence the aerosols only impact the radiation, not the microphysics. We chose to use the aerosol data from CAMS (Inness et al., 2019a) for this purpose, as it is also used e.g. in IFS

(Bozzo et al., 2017) and to study aerosol-radiation interactions in Witthuhn et al. (2021). The dataset includes information for 11 different aerosol types, namely organic matter (hydrophilic and hydrophobic), black carbon (hydrophilic and hydrophobic), sea salt (in 3 size ranges), dust (in 3 size ranges), and sulphates.

From the CAMS dataset, we used the aerosol mass mixing ratios at the CAMS model levels, which we obtained with the same workflow as we used to include the ERA5 meteorology (van Stratum et al., 2023). Thus, we downloaded the vertical 180 profiles based on a given location (lat/lon), after which we interpolated the data to both the LES vertical levels and the ERA5 model levels, the latter being necessary for the calculations of the radiation above the domain as described in Sect. 2.2.1. The CAMS aerosol mass maxing ratios have a temporal resolution of 3 hours and we linearly interpolate them in our simulations to the radiation timestep after which they are converted to optical properties using a pre-calculated lookup table with aerosol optical properties (Bozzo et al., 2020). The optical properties in this table are given per class of 10% relative humidity between 185 0 and 80% relative humidity and per class of 5% between 80% and 100% relative humidity. We used these optical properties directly, without interpolation between the classes. For our model domain, we combined one profile of aerosol mass mixing ratios with the 3D relative humidity in our simulation, resulting in aerosol optical properties that also varied in the horizontal. For our background profile, we used the relative humidity from ERA5. Finally, the aerosol optical properties were combined with the optical properties of the gases and the clouds, to form the total set of optical properties that was used for the (coupled 190 and uncoupled) radiation calculations.

## 3   Validation

We compare our simulations with observations from the Royal Netherlands Meteorological Institute (KNMI) site at the Ruisdael Observatory in Cabauw, the Netherlands. First, we validate our implementation of the aerosols for the clear sky days using the observations from the BSRN station (Mol et al., 2023). Next, we validate the cumulus cases, for which we use the 195 temperature, specific humidity, and wind measurements from the measurement tower (KNMI Data Services, 2024b) and the cloud cover measurements of the Nubiscope, which is a scanning infrared radiometer (KNMI Data Services, 2024a).

For the clear sky days, we compare the global, direct, and diffuse radiation of our simulations with the observations in Fig. 2. In the simulations without aerosols (top row), we notice that the simulated direct radiation is too high compared to the observations and the diffuse radiation is too low, because the scattering effect of the aerosols is missing. Also, we see that the 200 global radiation is slightly overestimated in the simulations compared to the observations, as the aerosols also absorb some radiation and increase the scattering back upwards. In the simulations with aerosols (bottom row), the direct, diffuse, and global radiation are well in line with the observations. This demonstrates that we can use the aerosol data from CAMS to remove a mean bias in the global, direct, and diffuse radiation.

For the cumulus days, we compare the observed and simulated cloud cover in Fig. 3. The selected days cover a range of 205 cumulus cloud conditions with the maximum simulated cloud cover between 0.2 and 0.75. Some differences between the observed and simulated cloud cover are expected for multiple reasons. Firstly, the simulated cloud cover is defined as the fraction of model columns that contain any liquid water, whereas the observed cloud cover includes part of the cloud sides

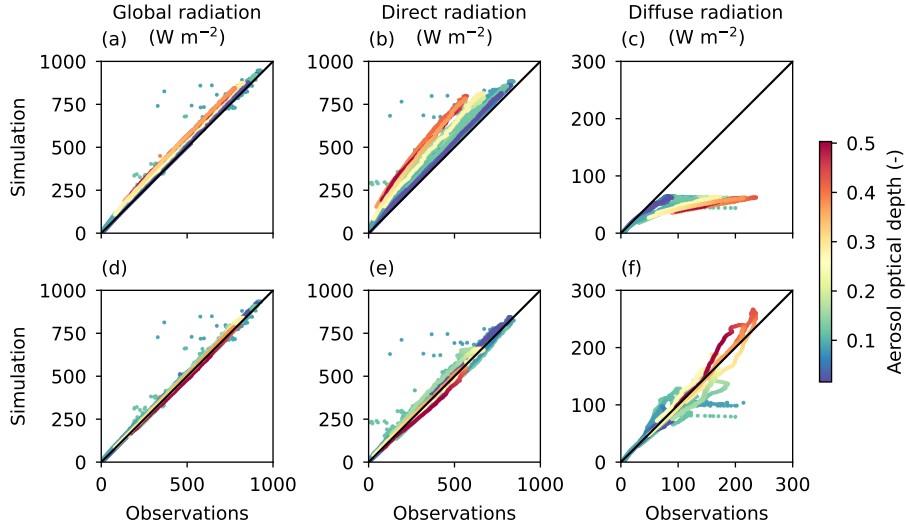

**Figure 2.** Comparison of simulated and observed surface solar radiation on clear sky days. The top row shows simulations without aerosols, the bottom row simulations with aerosols.

seen by the scanning instrument. We shortly investigated for one of our cases how sensitive our modelled cloud cover is to the chosen definition. Different definitions change the cloud cover but the differences were limited to a maximum of 0.06.
Hence, we chose to use the described model definition in the remainder of this paper. Secondly, the observed and simulated cloud cover differ as we likely miss part of the variability in cloud cover because of the limited domain size and double-periodic boundaries of our simulations, which prohibit the formation of meso-scale structures (e.g., Schalkwijk et al., 2015; Heinze et al., 2017; Schemann et al., 2020; van Stratum et al., 2023). Thirdly, it is known, e.g. from the Radiative-Convective Equilbrium Model Intercomparison Project (RCEMIP) (Wing et al., 2020) and from the Atmospheric Radiation Measurement
(ARM) intercomparison of shallow cumulus over land (Brown et al., 2002), that simulated clouds depend on many model aspects such as the choice of advection scheme, microphysics scheme and resolution. These model aspects potentially have a stronger influence on the cloud cover than 3D radiation, as 3D radiation has a limited influence on the cloud cover, which is shown in Fig. 3 and discussed in section 4.1. However, it is beyond the scope of this paper to examine the differences between observed and simulated cloud cover any further. Here, we aim to show that despite the differences and uncertainties,
the simulated and observed cloud cover are roughly in line. This shows that our simulations realistically represent these days, which makes the simulations suitable to investigate the impact of 3D radiation. Validation plots of the temperature, specific humidity, and wind speed are included in the Appendix. For these variables, there is good agreement between the simulations and the observations, which confirms that our simulations realistically represent these days.

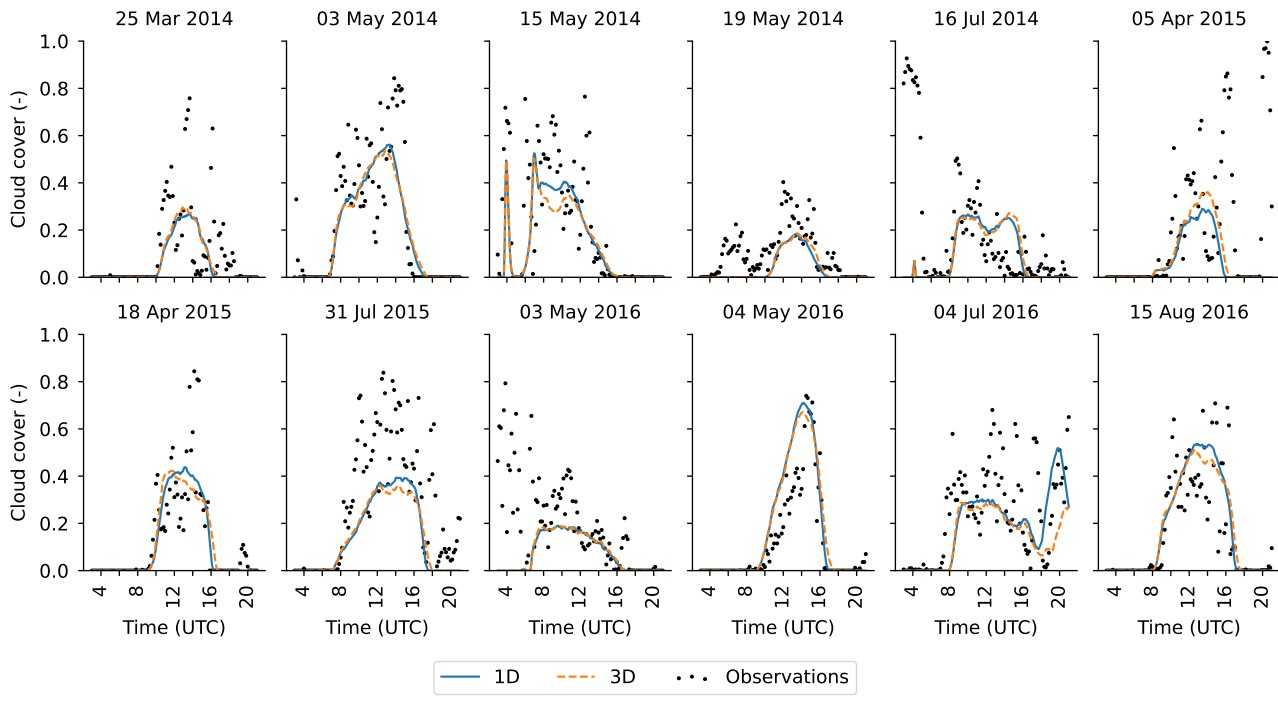

**Figure 3.** Time series of simulated and observed cloud cover on the 12 selected cumulus cloud days.

## 4 Results

We first compare the clouds in the simulations with 1D and 3D radiation in Sect. 4.1, after which we investigate the differences in domain-averaged surface radiation in Sect. 4.2.

### 4.1 Changes in clouds

Figures 4a, b, and c show the cloud cover, cloud depth, and liquid water path of the simulations with 1D and 3D radiation. The cloud cover is generally similar in the simulations with 1D and 3D radiation, but the cloud depth (defined as the distance between the lowest and highest model level with any liquid water) and liquid water path clearly differ. Simulations with 3D radiation have deeper clouds and a higher liquid water path than simulations with 1D radiation. The higher liquid water path is partly because of the deeper clouds, although the additional layers with liquid water in the simulations with 3D radiation contain relatively little liquid water, so the difference in liquid water path is mainly because of higher liquid water concentrations.

Furthermore, we investigate the spectra of vertical wind in the boundary layer and specific humidity in the cloud layer. These spectra give insight in the size of the turbulent structures that ultimately determine the cloud sizes. To summarize and compare the spectral information, we define the characteristic length scale of the spectra, as in Veerman et al. (2020) (and references therein). This length scale is shown in Fig. 4d for the specific humidity in the cloud layer, where the development of the length

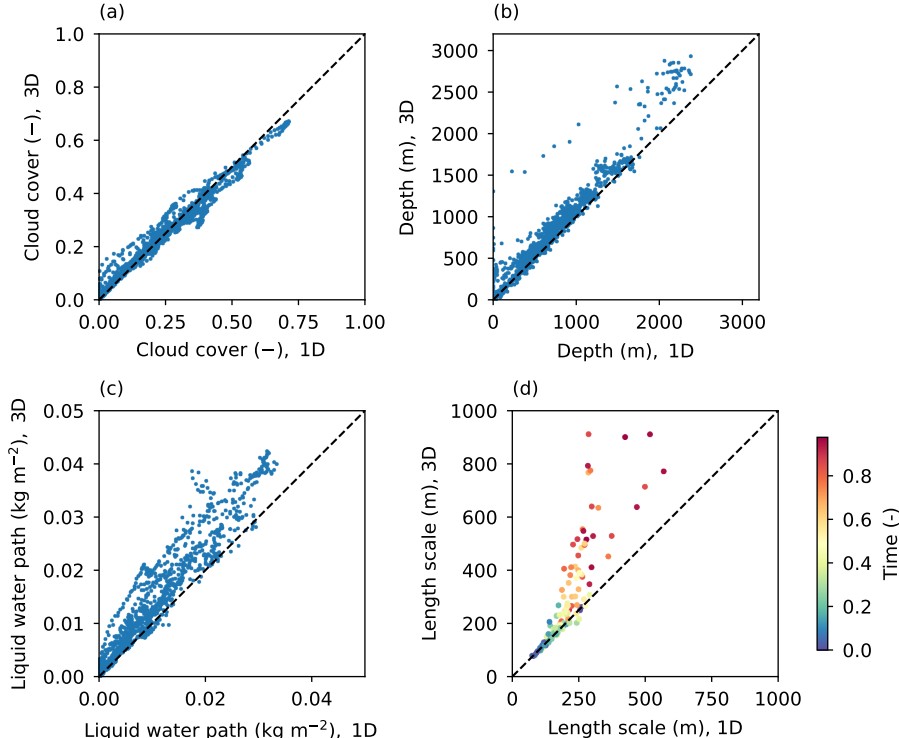

**Figure 4.** Comparison of the domain-averaged cloud characteristics of the simulations with 1D radiation and 3D radiation. (a) cloud cover, (b) cloud depth, (c) liquid water path, (d) characteristic length scale of specific humidity in the cloud layer. The length scales are determined from one simulation per radiation type instead of being the average of three simulations. The colors in (d) indicate the normalized time between the first and last timestep with any liquid water.

scale during the day can be seen from the colors that indicate the normalized time between the first and last time step with any liquid water. The characteristic length scale increases during the day in both simulations, as the clouds develop during the day. However, in the simulations with 3D radiation, the characteristic length scale increases more than in simulations with 1D radiation. We find a similar pattern for the characteristic length scale of the vertical velocity spectrum in the boundary layer (between 450 and 550 m, not shown), with larger length scales in the simulations with coupled 3D radiation. These increases in the length scales of vertical velocity and specific humidity show that the turbulent structures become larger, which results in larger clouds.

Our findings generalize the results of Veerman et al. (2020), who found larger and thicker clouds, and Veerman et al. (2022), who found an increase in liquid water path, wider clouds, and a similar cloud cover. Thus, simulations with 3D radiation develop deeper and wider clouds with a higher liquid water path but similar cloud cover as simulations with 1D radiation.

Apart from the differences in cloud properties found by Veerman et al. (2020, 2022) and generalized here, Jakub and Mayer (2017) found that coupled 3D radiative transfer can influence the cloud organization. Unfortunately, their quantification of

250 the cloud organization only describes organization in the north-south and east-west direction, which captures the cloud streets in their idealized setup but not in our complex cases. Further investigation of the cloud organization with other measures is beyond the scope of this paper.

As described in Sect. 2, the two-way feedback between clouds and surface radiation obscures the origin of deeper and wider clouds under similar cloud cover, but the variation among our cases can shed some light on which changes in the surface

radiation matter for cloud development. To this end, we focus on the correlation between the cloud shadow displacement and the changes in liquid water path. We find that the changes in liquid water path are significantly correlated to the wind-sun angle, the wind speed and the horizontal distance between a cloud and its shadow. The highest correlation is found between the relative difference in liquid water path and the wind-sun angle (r = 0.55). If the wind-sun angle is small, a cloud moves in the direction of its own shadow, where the surface fluxes are reduced. This suppresses the formation of updrafts, and is

therefore disadvantageous for the growth of the cloud. Hence, the clouds in the 3D simulations cannot grow bigger than in the simulations with 1D radiation and the relative difference in liquid water path is small. In contrast, if the wind-sun angle is large, the cloud does not travel over its own shadow, but instead over a sunlit area that potentially receives additional radiation through cloud enhancements. The updrafts likely form in these sunlit areas and the cloud will be enforced by these updrafts, causing the cloud to live longer and therefore potentially grow more than in a simulation with 1D radiation. Hence, the relative

difference in liquid water path is large. Therefore, the correlation between the wind-sun angle and the relative difference in liquid water path supports the hypothesis that the 3D effects propagate via the surface.

The other investigated factors, wind speed and the horizontal distance between a cloud and its shadow, are also positively correlated with the relative difference in liquid water path, although the correlations are weaker (r = 0.33 and r = 0.27, respectively). As our three factors are independent of each other, we perform a multilinear regression that combines these three

factors. This combination has a slightly improved correlation compared to the wind-sun-angle alone (r = 0.6). This increased correlation shows that the wind speed and the horizontal distance between a cloud and its shadow also play a role, but their impact is not as clear as for the wind-sun angle. Furthermore, we note that the three factors combined do not fully explain the relative difference in liquid water path. This highlights the complexity of the problem, where clouds can also be influenced by the shadows of other clouds and by other factors that control cloud formation, such as the stability of the layer above the clouds

or the soil moisture content.

## 4.2 Changes in radiation

In this section, we investigate the differences in domain-averaged surface radiation. We first describe the coupled effect, so radiation from the simulations with 3D radiation minus radiation from the simulations with 1D radiation. Then, we describe and explain the radiation effect and cloud effect. Thereafter we explain the coupled effect using the radiation effect and cloud

effect. For each effect, we show time series of the effect for one arbitrarily selected example day (3 May 2014, Fig. 5) and box plots including all days (Fig. 6). We end this section with a short note about the upward radiation at the top of the domain, which is therefore included in (Fig. 6).

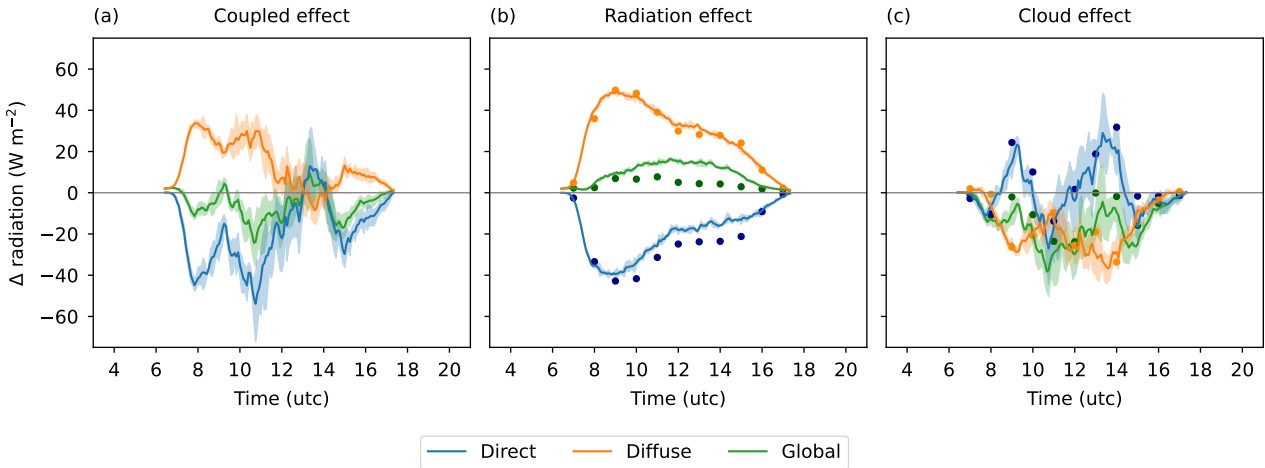

**Figure 5.** Time series of the coupled effect (a), radiation effect (b), and cloud effect (c) on 3 May 2014. In (b) and (c), the dots show the effect when using uncoupled 3D radiation ($1D_{rad3D}$) and the lines show the effect when using uncoupled 1D radiation ($3D_{rad1D}$). The shading shows the range of the differences between the repetitions of the simulations.

Figure 5a shows the mean differences in radiation between the simulations with 3D radiation and the simulations with 1D radiation on one day. On this day, increases and decreases in global, direct, and diffuse radiation all occur when using coupled 3D radiation instead of coupled 1D radiation. Also shown in the figure is the spread between the different simulation realizations of the selected day. As three simulations with different random seed for the initial random perturbation for both the simulations with 1D and 3D radiation, the cloud effect and coupled effect can be calculated nine times and the spreading shows the minimum and maximum of these nine possible combinations. The radiation effect does not combine simulations with 1D and 3D radiation, hence the spreading shows the minimum and maximum of three simulations. The spreading illustrates that the trends and sign are the same across the repeated simulations, although the exact magnitude of the effects differs.

The mean coupled effect for all days is summarized in the box plots in Fig. 6a. On average, the direct radiation decreases in the simulations with 3D radiation compared to the simulations with 1D radiation, the diffuse radiation increases, and the global radiation stays approximately the same. However, individual moments can deviate strongly from this general pattern, for example the difference in global radiation can be more than +/- 25 W m$^{-2}$ instantaneously.

To understand the coupled effect better, we separate it in the radiation effect and the cloud effect, which are shown for one day in Fig. 5b and c. As explained in Sect. 2, we can split the coupled effect in two ways. We can use the uncoupled 1D radiation ($3D_{rad1D}$, top right in Fig. 1), which results in the lines in Fig. 5b and c. Alternatively, we can use the uncoupled 3D radiation ($1D_{rad3D}$, bottom left in Fig. 1), which results in the dots in Fig. 5b and c. Comparing the two splitting methods shows that the exact magnitude of the effects depends on the splitting method, which is not surprising for the day in Fig. 5 given the large difference in domain-averaged liquid water path (up to 0.012 kg m$^{-2}$) between the simulation with 1D and 3D radiation. However, the trends and sign of the effects are the same between the splitting methods, thus we can use either method to explain

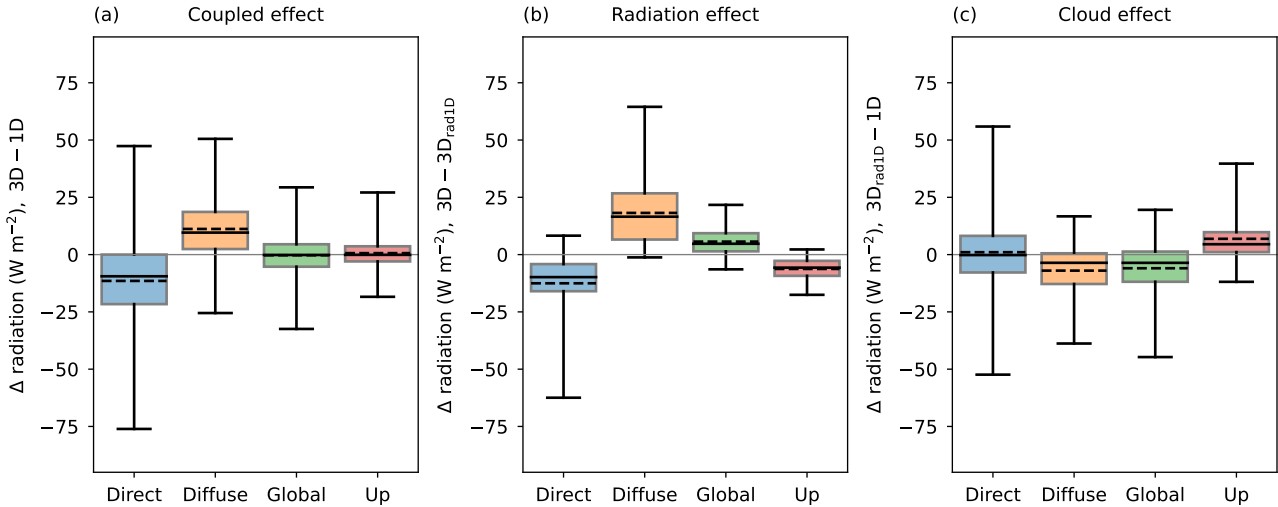

**Figure 6.** Box plot of the coupled effect (a), radiation effect (b), and cloud effect (c) in direct, diffuse and global radiation at the surface and upward radiation at the top of the domain. The box plots include all times with clouds on the twelve selected cumulus days. The whiskers range from the minimum to the maximum difference, the full line indicates the median, and the dashed line shows the mean.

them. We opt for the splitting using uncoupled 1D radiation, as this splitting method has the advantage that we compare two versions of 1D radiation for the cloud effect, which is the simplest to understand, and the uncoupled 1D radiation is available at higher frequency than the uncoupled 3D radiation (see Fig. 5).

### 4.2.1 The radiation effect

First, we look at the radiation effect, which is the difference between 1D and 3D radiation for the same clouds. Figure 5b shows the radiation effect for one day, but we find a similar pattern for the other 11 days (not shown). Figure 6b shows box plots of the radiation effect using uncoupled 1D radiation for all cases together. Both plots show that for a given cloud field the direct radiation generally decreases with 3D radiation compared to 1D radiation, and both the diffuse radiation and global radiation generally increase. These changes are in line with the results of Gristey et al. (2020a), where they also explain how these results are the combination of side-escape, side-illumination, and entrapment. In short, diffuse radiation increases because of the side escape and entrapment, whereas direct radiation decreases because of side illumination. Gristey et al. (2020a) also found that the difference in direct radiation can sometimes be positive. We see this as well for a short moment on one day. In our case, this is because the clouds are tilted along the angle of the incoming sunlight. In addition, Gristey et al. (2020a) found that the difference in global radiation can be slightly negative at the end of the day as the side-illumination effect dominates at large solar zenith angles. We see the same effect on a couple of days, which explains the few negative values that can be seen in Fig. 6b. In summary, for our cases (and also for the ones in Gristey et al. (2020a)), the radiation effect is an increase in diffuse radiation and a decrease in direct radiation, which combined results in a net increase in global radiation.

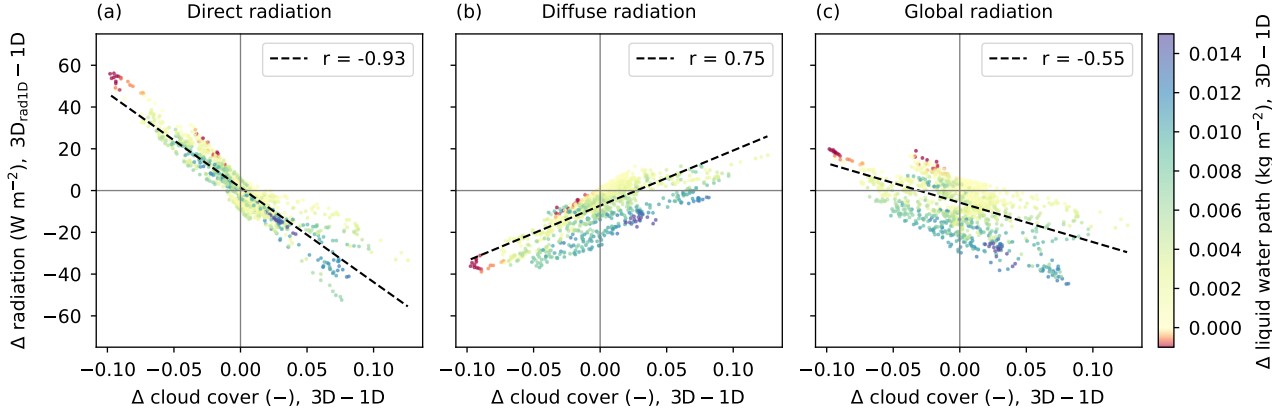

**Figure 7.** The cloud effect as a function of the difference in cloud cover. (a) Direct radiation, (b) diffuse radiation, and (c) global radiation. The colors indicate the difference in liquid water path. The black dotted line shows the linear trend with the correlation coefficient (r).

### 4.2.2 The cloud effect

Next, we look at the cloud effect, which can be seen as a difference in radiation caused by a change in clouds. Since we use the uncoupled 1D radiation, we are looking at the difference in 1D radiation between two simulations with different clouds (namely the simulation with coupled 3D radiation and the simulation with coupled 1D radiation). This difference is indicated with the right green line in Fig. 1 ($3D_{rad1D}$ - 1D). The lines in Fig. 5c show this cloud effect for one day, and Fig. 6c shows box plots of this cloud effect for all cases together.

For the largest part of the day in Fig. 5, the direct radiation is higher when calculating uncoupled 1D radiation of a simulation with coupled 3D radiation ($3D_{rad1D}$) compared to the simulations with coupled 1D radiation. However, for a short period, there is a decrease. Conversely, both the diffuse and global radiation decrease. Looking at all 12 days, there is no clear daily pattern (not shown). Figure 6c reveals that increases and decreases in direct radiation occur roughly equally much, but typically both the diffuse and global radiation decrease. Thus, in contrast to the radiation effect, the cloud effect is a net decrease in global

radiation.

   The changes in direct and diffuse radiation, and therefore the net decrease in global radiation, are correlated with the changes in clouds. In general, the direct radiation at the surface is minimal in cloud shadows and high in clear sky areas. Consequently, a change in direct radiation is mainly caused by differences in shadowed area. As we are comparing 1D radiation of different cloud fields, a difference in shadowed area means a difference in cloud cover. Figure 7a confirms that the differences in direct

radiation are strongly anti-correlated with the differences in the cloud cover, and that there is no difference in direct radiation when the cloud cover is the same in the simulations with 3D and 1D radiation.

   The opposite reasoning holds for the differences in diffuse radiation, which is shown in Fig. 7b. The diffuse radiation increases when the cloud cover is higher in the simulation with 3D radiation compared to the simulation with 1D radiation. However, the differences in diffuse radiation are not only caused by differences in cloud cover, but also by differences in liquid

water path, as can be seen from the colors in Fig. 7b. When the liquid water path is higher, there is less diffuse radiation at the surface because there is more absorption and more scattering back upwards.

    Combining the cloud effect on direct and diffuse radiation results in the cloud effect in global radiation shown in Fig. 7c. This effect is not as strongly correlated with the difference in cloud cover, because the direct and diffuse effect partly cancel each other out. Apart from more radiation being scattered downwards, an increased cloud cover also increases upward scattering

and absorption, which reduces the global radiation at the surface. Therefore, the difference in global radiation is negatively correlated with the difference in cloud cover. Similar to the difference in diffuse radiation, the difference in global radiation is also related to the difference in liquid water path, as a larger increase in liquid water path corresponds to a larger decrease in global radiation.

### 4.2.3   The coupled effect

The radiation effect and the cloud effect together give the coupled effect from the beginning of this section: a decrease in direct radiation, an increase in diffuse radiation, and no change in global radiation. We can now explain this coupled effect, using the radiation effect and cloud effect.

    Figure 8 shows the correlation between the changes in global radiation and the changes in clouds. Similar to what we see for the cloud effect, the changes in global radiation and cloud cover are negatively correlated (Fig. 8a). The trend line shows

that there is no difference in global radiation when there is no difference in cloud cover, which is in line with what we find on average for our 12 days. Comparing Fig. 8a and Fig. 7c reveals that the coupled effect in global radiation is largely comparable to the cloud effect in global radiation. The strong similarity between these figures emphasizes how important the changes in clouds are for the coupled effect.

    We can also recognise the uncoupled effect in the coupled effect from the correlation between the differences in global

radiation and the differences in liquid water path (Fig. 8b). When the difference in liquid water path (and cloud cover) is close to zero, the global radiation is higher in simulations with coupled 3D radiation. Thus, when the clouds are the same, 3D radiation gives more global radiation, which is also what we find as the uncoupled effect. Following a similar reasoning, Figure 8b shows that the global radiation is the same in the simulations with 1D and 3D radiation, when the simulation with 3D radiation has an increased liquid water path, which is exactly what we find on average for our 12 days.

**4.2.4   Changes at the top of the domain**

    We end this section by shortly looking into the differences in radiation at the top of our domain. Figure 6 shows that the radiation effect is a decrease in upward shortwave radiation and the cloud effect is an increase in upward shortwave radiation at the top of the domain. In section 4.2.2 we wrote that the increase in liquid water path reduces the surface radiation as absorption and scattering back upwards are increased. More in general, any increase (decrease) in radiation at the surface has

to be compensated by reduced (enhanced) scattering back upwards or reduced (enhanced) absorption, as the incoming radiation is the same. The box plots show that the differences at the top of the domain are roughly the opposite of the differences in global radiation at the surface, hence the differences in absorption, that cause differences in heating, are limited when averaged

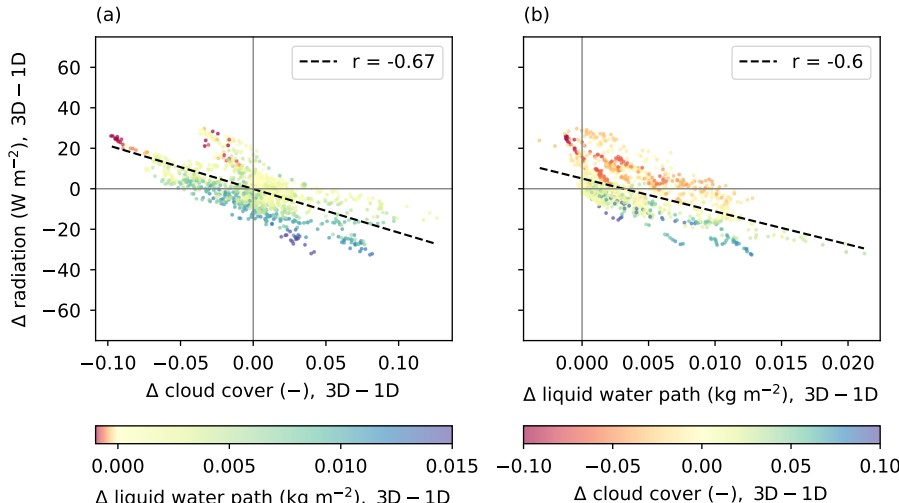

**Figure 8.** The coupled effect in global radiation as a function of the difference in cloud cover (a) and liquid water path (b). The colors indicate the difference in liquid water path (a) or cloud cover (b). The black dotted line shows the linear trend with the correlation coefficient (r).

over the entire domain. Similar to the coupled effect in global radiation at the surface, the coupled effect at the top of the domain is close to zero because of the opposing impacts of the radiation effect and cloud effect.

## 5 Conclusions and outlook

We studied the impact of 3D radiation on cumulus clouds and the domain-averaged surface radiation, by comparing coupled 1D and coupled 3D radiation for 12 cumulus cases. We found that coupled 3D radiation increases cloud liquid water path and cloud size (both in the horizontal and in the vertical), but without affecting the cloud cover. The domain-averaged surface global radiation is also on average unchanged because of two opposing effects. On the one hand, uncoupled 3D radiation causes a decrease in direct radiation by side illumination and an increase in diffuse radiation by side escape and entrapment, resulting in a net increase in global radiation. On the other hand, an increase in liquid water path of the clouds causes a decrease in global radiation.

Our results show that the necessity for coupled 3D radiation depends on your goals. For example, the increased cloud size and liquid water path might feed back to rain formation, which is relevant for weather prediction applications. Furthermore, the shift in the partitioning between direct and diffuse radiation can influence photosynthesis and energy production by solar panels. In addition, the current results might be used in future research to validate parameterizations of 3D effects when these parameterizations are coupled to simulations.

Future work could tackle the limitations of the current work, to generalize our results further. For example, the coupled ray tracer could be extended to the longwave spectral range, as previous research has shown that 3D longwave radiation also influences the clouds (see e.g. Schäfer et al. (2016); Klinger et al. (2017)). In addition, our clouds are currently bound by

the limited domain size and periodic lateral boundaries. To determine if and how this influences the impact of 3D radiation we recommend a setup with open boundaries for future research. Moreover, it is not trivial to determine from our current simulations which factors determine the magnitude of the differences between simulations with 1D and 3D radiation. To this end, a more idealistic setup, in which one factor is changed at a time, might yield further insights. Lastly, the current work focuses on cumulus clouds over grassland in the mid-latitudes. Although cumulus clouds in other regions can have similar cloud properties (Dror et al., 2020), the impact of 3D radiation might be different because of difference solar zenith angles. Moreover, the impact of coupled 3D radiation is potentially different for other cloud types, where the coupling with the surface is less important.

Nonetheless, we believe that our 12 cases cover the most common conditions with cumulus over grassland in the mid-latitudes and that our results are representative for those conditions. Hence, we conclude that, coupled 3D radiation deepens cumulus clouds without changing the mean surface solar irradiance.

*Code and data availability.* The observations of temperature, humidity, wind and cloudcover at the measurement station in Cabauw are openly available from the KNMI Data Platform (KNMI Data Services, 2024a, b). The observations of radiation and the additional data used to select the cases are openly available from Knap and Mol (2022) and Mol et al. (2022). The CAMS global reanalysis data is openly available from Inness et al. (2019b) and Inness et al. (2019c). The model simulations are performed with MicroHH (van Heerwaarden et al., 2017) version 2.0.0_RC1, which is openly available at https://github.com/microhh/microhh/releases/tag/2.0.0_RC1, coupled to ERA5 using $(LS)^2D$ (van Stratum et al., 2023), which is openly available at https://github.com/LS2D/LS2D and https://pypi.org/project/ls2d/. The complete model set-up of our simulations as well as scripts to analyse the simulation results can be found at https://doi.org/10.5281/zenodo.11234716

## Appendix A

Figure A1 shows the observed and simulated temperature, specific humidity and wind speed.

*Author contributions.* MT performed the simulations and analysis and wrote the manuscript in close collaboration with BvS and CvH.

*Competing interests.* The authors declare that they have no conflict of interest.

*Acknowledgements.* We acknowledge funding from the Wageningen Institute for Environment and Climate Research (WIMEK) and the Shedding Light On Cloud Shadows project funded by the Dutch Research Council (NWO) (grant: VI.Vidi.192.068). The simulations are carried out on the Dutch national e-infrastructure with the support of SURF Cooperative (project numbers NWO-2021.036 and NWO-2023.003).

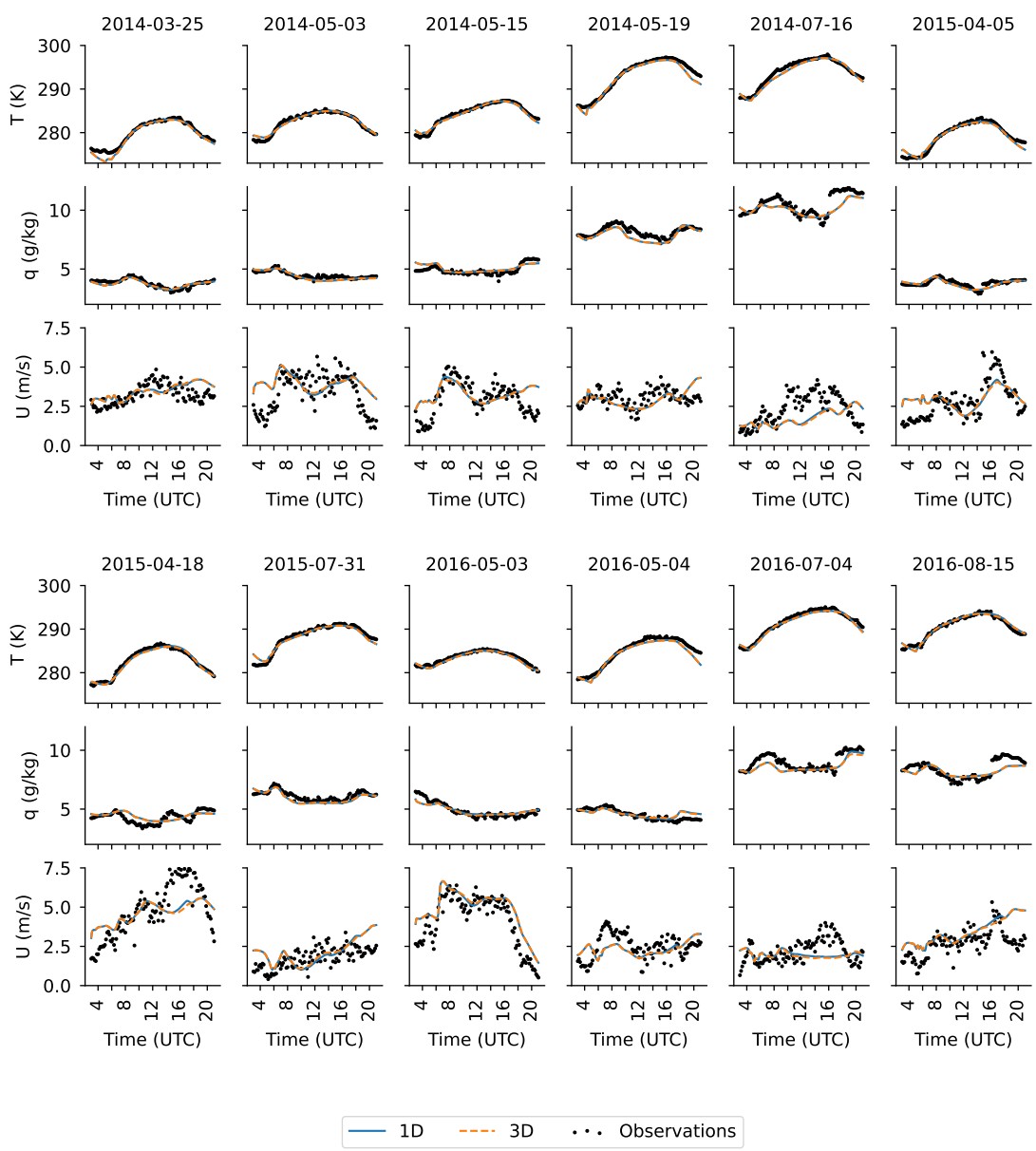

**Figure A1.** Time series of simulated and observed temperature (first and fourth row), specific humidity (second and fifth row), and wind speed (third and sixth row) on the 12 selected cumulus cloud days. Observations are at 10 m height and values from the simulation are taken at 12.5 m height.

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
