# Peer review of "The impact of coupled 3D shortwave radiative transfer on surface radiation and cumulus clouds over land"

_EGUsphere, 2024_

## Referee Comment (RC1)

In this manuscript, the authors couple a 3D ray-tracing radiative transfer scheme to large eddy simulations of cumulus clouds in order to investigate the effect of 3D radiative transfer, including feedbacks on cloud properties due to instantaneous changes in radiative fluxes during cloud development. This kind of study is important, because as the authors demonstrate, the coupled effect in global radiation is largely due to the changes in clouds ("the cloud effect"). They found that coupled 3D radiation increases cloud liquid water path and cloud size, which causes a decrease in global radiation, and to a certain extent, this counterbalances the increase in global radiation due to uncoupled 3D radiative transfer. Even though the authors found a net difference in global radiation of only $1 \text{ W m}^{-2}$ on account of this counterbalancing effect, they revealed the importance of the changes in the clouds themselves due to the coupling, and the assessment of the magnitude of the effect is important in and of itself. The manuscript is well written, and the results are interesting and important. As such, in my opinion, this manuscript is appropriate for publication in Atmospheric Chemistry and Physics pending some clarifications and small corrections, as I list below.

1.  lines 15-16: There are a few other works that could be cited here: Marshak, A. and A. Davis, Eds., *3D Radiative Transfer in Cloudy Atmospheres*, Springer, 2005, and references therein; additional research by R. Pincus not already cited.
2.  lines 44-51: In previous studies, which specific changes in cloud development were found to be caused by shortwave 3D radiative transfer, such as cloud shadows, and which specific effects on cloud development were found to be caused by longwave 3D radiative transfer? If the authors do not want to explain that here, then perhaps refer the reader here to section 2.2.2.
3.  lines 56-58: "To validate the inclusion of aerosols, we performed simulations for a set of days with clear skies over Cabauw, the Netherlands, which we compared with observations." – I suggest adding "see section 3" here.
4.  lines 58-59: "Next, we used the setup with aerosols to simulate a set of 12 days during which shallow cumulus clouds developed. After comparing the results with observations to ensure that the simulations resemble reality…" – Are these sentences referring to simulations with or without coupled 3D radiative transfer, or both?
5.  line 133: "The impact of coupled 3D radiation, hereafter referred to as the coupled effect, is the difference between the two, so 3D – 1D," – This is a little confusing at first, because it sounds like the impact of the dimensionality, not an isolation of the coupling, but the distinction becomes clearer in the sentences that follow.
6.  lines 142-143: "The uncoupled effect using uncoupled 3D radiation was studied before, e.g. by Gristey et al. (2020a)." – Similar comment to my comment 1 above.
7.  line 158: "based on a give location" – "based on a given location"
8.  line 185: "However, we miss part of the variability in cloud cover, which is likely because of the limited domain size and double-periodic boundaries of our simulations, which prohibit the formation of meso-scale structures…" – The authors refer to this fact again when they discuss future studies in section 5, but they should also mention whether this fact could have influenced the magnitude and direction of their results.

9.  lines 194-195: "Simulation with 3D radiation have deeper clouds" – "Simulations with 3D radiation have deeper clouds"

10. lines 309-10, 310-311, and 318-319: Do these statements conflict one another?
    "When the difference in liquid water path (and cloud cover) is close to zero, *the global radiation is higher* in simulations with coupled 3D radiation."
    and
    "… when the clouds are the same, 3D radiation gives *less global radiation*, which is also what we find as the uncoupled effect…"
    and
    "… uncoupled 3D radiation causes a decrease in direct radiation by side illumination and an increase in diffuse radiation by side escape and entrapment, resulting in a *net increase in global radiation*…"

---

## Referee Comment (RC3)

**Review of "The impact of coupled 3D radiative transfer on surface radiation and cumulus clouds over land" by Tijhuis et al.**

24 June 2024

*General comments*

This paper presents large eddy simulations of continental cumulus clouds with a coupled shortwave 3D radiation scheme. Results are compared to simulations with a standard 1D radiation scheme to assess the impact of shortwave 3D radiation effects on the clouds and surface radiation. Simulations with 3D radiation are found to have larger and deeper clouds, which reflect more shortwave radiation and therefore reduce the surface shortwave radiation. This acts in the opposite direction to non-coupled 3D radiation effects that tend to enhance the surface shortwave radiation. Overall, there is an almost exact compensation such that the mean surface shortwave radiation in simulations with 1D and 3D radiation is very similar. This finding is based on multiple simulated days.

The paper is clear and well written. I learned a lot from reading this paper. I would like to congratulate the authors on their significant new findings. The conclusions are mostly convincing, and not necessarily expected. I think this paper will spur further studies and will become a well cited reference in the coming years. I have a few suggestions to further strengthen the study, as outlined in my comments below. After addressing these comments, I believe the study will be appropriate for publication in ACP.

*Specific comments*

Title: I suggest changing to "The impact of coupled shortwave 3D radiative transfer on surface radiation and cumulus clouds over land" because the authors did not consider coupled longwave. This is already mentioned in the manuscript, but it would add clarity to put this extra word in the title.

Introduction: Several references are made to "cloud resolving models". I think the authors are only referring to large eddy simulations (LES). Often, the term "cloud resolving model" is used to refer to a model that is run without a convection scheme. These models can still have horizontal resolution of a few km, but I don't think that's what the authors intend. I recommend saying LES from the beginning, or even better explicitly state the range of horizontal resolutions that are used in the cited studies.

L47-49: It is interesting that the study from Jakub and Mayer claimed that cloud organization (cloud streets) occur with coupled 3D radiation, but the studies by Veerman et al. and the present study do not seem to identify or even mention cloud organization. I think it is worth mentioning this difference in the introduction and/or conclusions as a potential discrepancy that exists in the literature.

L75-76: It would be helpful to expand on what is meant by "the days where the simulated cloud cover visually matches the observed cloud cover". What is the criteria for a match? For example, does the match consider only cloud cover or also cloud size, shape, organization, etc?

Section 2.1: I recommend listing the exact dates that were used for the clear-sky (13 days), cloudy (20 days), and subset cloudy (12 days). Or put them in a table. This is needed for reproducibility reasons.

L82: I do not see any supplementary materials uploaded. Can the details of the LES model be referred to previous literature?

L88: A couple of sentences explaining what the land surface model is actually doing would help here. How realistic is the assumption of an instantaneous surface response? I am concerned that this assumption might make the coupling with the clouds too strong. If, in reality, there is some delay of the surface response, then the clouds could evolve or be advected before they "feel" the surface immediately below them. I would have thought that the land surface model needs to account for the processes that determine the response time of the surface such as heat transfer between vegetation and soil layers, and stomatal opening/closing.

L90: Can some justification be provided that the radiation calls once every minute are sufficient? The appendix shows that the wind speed often exceeds 5 m/s at 10 m altitude. The wind speed at cloud altitude is probably even larger. Taking 5 m/s as a typical wind speed, in the 1 minute between radiation calls a cloud would be advected 300 m. Figure 4d shows that 300 m is comparable to the length scale of the simulations. This means that an individual cloud will move a distance that is comparable to the size of its shadow before the position of the shadow is updated in the simulation, which seems quite "jumpy". Given that the study depends critically upon the land-atmosphere coupling imposed by the evolving pattern of cloud shadows, I think the reader needs some more convincing on this decision.

L108-112: This hypothesis would benefit from some discussion of the time/length scales of the boundary layer mixing. The surface is not instantaneously connected to the cloud immediately above it. It takes time for perturbations in surface fluxes to be transported up to cloud base. And during this transport there must also be some mixing that occurs, such that the variability at cloud base is not as sharp as the surface. For the clouds to be influenced, I think it has to be the case that: 1. the timescale for mixing to cloud base is shorter than the timescales of individual cloud evolution and movement, and 2. the surface discontinuities are not simply mixed away during transport to the cloud layer. For example, one recent study that considered how these types of clouds change during a solar eclipse suggested that the fastest timescale for surface air parcels to be transported to cloud altitude is around 15 minutes (https://doi.org/10.1038/s43247-024-01213-0). It is not clear that this is shorter than the timescales of cloud movement and growth/decay, which leads me to question this hypothesis.

L158: Give -> Given

L161-162: At large RH, above 90% or so, even a small change in RH can result in a large change in optical properties due to the non-linearity in aerosol extinction as a function of RH. Are the optical properties defined per 10% RH even at high RH? If so, this likely introduces an important source of error for aerosol optical properties in the vicinity of clouds. These errors will not be evident in the clear sky cases that are used for validation, but they will be present on cloudy days because RH will approach 100% toward cloud edges.

L172: It should be noted that the definition of cloud cover in observations and models is slightly different here. A scanning or wide-field view instrument will detect and include cloud sides as part of the overall cloud cover. In contrast, cloud cover in the LES model is defined only from a zenith view perspective. This will generally lead to an overestimate of cloud cover in observations relative to LES, unless an instrument simulator is used within the LES to ensure sampling consistency (which I don't think is done here). This fact also has implications for one of the main results of the study, that the clouds are deeper in 3D coupled radiation but the cloud cover is the same. This may be true with the model definition of cloud cover. But from an observational perspective, the cloud cover could actually

still increase because the deepening of clouds results in more of the sky becoming obscured at oblique views of the instrument. This is worth commenting on in the conclusions.

L211-223: Can a statistical significance test be done to determine whether the correlations presented are significant? That would make the presented correlations more convincing.

L211-231: Similar to my comment above about the hypothesis for cloud changes, I think this discussion would benefit from considering the time and space scales involved. If the hypothesis holds, the correlation should be highest for a combination of the factors explored: when the wind direction is aligned with the sun angle AND the shadow displacement divided by the wind speed is similar to the cloud base height divided by the updraft speed. Could the authors look at this explicitly and see if they find a connection? I also wonder if it is possible that this combination of factors could lead to a suppression of cloud development in 3D in the case that clouds are systematically moving toward their shadows in 3D. Do the authors see any evidence of this? If not, does this provide evidence to reject the proposed hypothesis?

Fig. 6: Are the box plots showing the mean across the entire day or at a specific time during the day? I might have missed it but I don't see this mentioned anywhere.

Section 4.2.1 and Fig. 5b: The global uncoupled effect is always positive, meaning that the diffuse effect dominates throughout the day. The Gristey et al paper that is already cited showed that the global effect can be negative at the end of the day on some days, because the direct effect can dominate at oblique sun angles. I am curious, do the authors see this on any of their simulated days, or is the global effect positive for all times of day and all cases? This would be an interesting similarity or difference to note in the paper.

Schematic figure of key result: I think this paper would really benefit from a schematic figure that captures the main result in the abstract ie. the almost exact compensation between uncoupled and cloud 3D effects. I encourage the authors to consider creating a schematic that represents this result in an intuitive and concise way. Figure 1 achieves this for the methodology. I am thinking of something similar to Figure 1, but for the results. This type of figure can help to engage a broader audience and increase the impact of the study.

---

## Author Comment (AC1)

review of egusphere-2024-1519

In this manuscript, the authors couple a 3D ray-tracing radiative transfer scheme to large eddy simulations of cumulus clouds in order to investigate the effect of 3D radiative transfer, including feedbacks on cloud properties due to instantaneous changes in radiative fluxes during cloud development. This kind of study is important, because as the authors demonstrate, the coupled effect in global radiation is largely due to the changes in clouds ("the cloud effect"). They found that coupled 3D radiation increases cloud liquid water path and cloud size, which causes a decrease in global radiation, and to a certain extent, this counterbalances the increase in global radiation due to uncoupled 3D radiative transfer. Even though the authors found a net difference in global radiation of only 1 W m−2 on account of this counterbalancing effect, they revealed the importance of the changes in the clouds themselves due to the coupling, and the assessment of the magnitude of the effect is important in and of itself. The manuscript is well written, and the results are interesting and important. As such, in my opinion, this manuscript is appropriate for publication in Atmospheric Chemistry and Physics pending some clarifications and small corrections, as I list below.

We want to thank the reviewer for the kind words about our manuscript and for taking the time to review our manuscript. We address the suggested clarifications and corrections below (in green text).

1.  lines 15-16: There are a few other works that could be cited here: Marshak, A. and A. Davis, Eds., 3D Radiative Transfer in Cloudy Atmospheres, Springer, 2005, and references therein; additional research by R. Pincus not already cited.
    We will add the suggested references.
2.  lines 44-51: In previous studies, which specific changes in cloud development were found to be caused by shortwave 3D radiative transfer, such as cloud shadows, and which specific effects on cloud development were found to be caused by longwave 3D radiative transfer? If the authors do not want to explain that here, then perhaps refer the reader here to section 2.2.2.
    We will elaborate the indicated paragraph such that it names the changes in cloud development that were found before.
3.  lines 56-58: "To validate the inclusion of aerosols, we performed simulations for a set of days with clear skies over Cabauw, the Netherlands, which we compared with observations." – I suggest adding "see section 3" here.
    We will add this.
4.  lines 58-59: "Next, we used the setup with aerosols to simulate a set of 12 days during which shallow cumulus clouds developed. After comparing the results with observations to ensure that the simulations resemble reality..." – Are these sentences referring to simulations with or without coupled 3D radiative transfer, or both?
    We will add here that these were simulations with 1D radiative transfer. However, as the cloud cover with coupled 3D radiative transfer is very similar to the cloud cover with 1D radiative transfer, we would have come to the same selection if we had used simulations with coupled 3D radiation or both.

5. line 133: "The impact of coupled 3D radiation, hereafter referred to as the coupled effect, is the difference between the two, so 3D – 1D," – This is a little confusing at first, because it sounds like the impact of the dimensionality, not an isolation of the coupling, but the distinction becomes clearer in the sentences that follow.
We thank the reviewer for pointing out that this is not immediately clear. We will rephrase the sentence such that it is immediately clear that we are referring to the schematic and the experiments as we labelled them for the schematic.

6. lines 142-143: "The uncoupled effect using uncoupled 3D radiation was studied before, e.g. by Gristey et al. (2020a)." – Similar comment to my comment 1 above.
We explicitly meant to refer to the paper by gristey et al here, as they have exactly the same effect in a way that is directly comparable to ours. We will formulate this more explicitly.

7. line 158: "based on a give location" – "based on a given location"
We will change this as suggested.

8. line 185: "However, we miss part of the variability in cloud cover, which is likely because of the limited domain size and double-periodic boundaries of our simulations, which prohibit the formation of meso-scale structures…" – The authors refer to this fact again when they discuss future studies in section 5, but they should also mention whether this fact could have influenced the magnitude and direction of their results.
We will mention in section 5 that this fact might have an influence on the results. Unfortunately, with the current setup it is impossible to tell if it impacts the results and if so what the magnitude and direction would be. It is not straightforward to reason what would happen, as cloud enhancements and the position of the cloud shadows relative to the cloud depends on much more than just the cloud size. One would need open boundary conditions to test this, which we have recently implemented in our model. We will start using this feature in future research to further investigate this.

9. lines 194-195: "Simulation with 3D radiation have deeper clouds" – "Simulations with 3D radiation have deeper clouds"
We will change this as suggested.

10. lines 309-10, 310-311, and 318-319: Do these statements conflict one another? "When the difference in liquid water path (and cloud cover) is close to zero, the global radiation is higher in simulations with coupled 3D radiation." And "… when the clouds are the same, 3D radiation gives less global radiation, which is also what we find as the uncoupled effect…" and "… uncoupled 3D radiation causes a decrease in direct radiation by side illumination and an increase in diffuse radiation by side escape and entrapment, resulting in a net increase in global radiation…"
We thank the reviewer for pointing this out and we apologise for these conflicting statements. We made a mistake in the second phrase mentioned. This should be more global radiation instead of less global radiation.

---

## Author Comment (AC2)

**Review of the manuscript "The impact of coupled 3D radiative transfer on surface radiation and cumulus clouds over land" by Tijhuis et al., 2024**

In this manuscript, the authors study the effects of 3D atmospheric radiation transfer (RT) on the surface's energy budget and cloud's properties for cumulus cloud fields over land. They analyze a unique dataset of a dozen Large eddy simulations (LES) with online calculations of 3D RT in the solar spectrum. This is unique since all LES simulations use 1D RT to decrease computational load. Here the issue is solved by using GPU for enhanced calculations.

The novel dataset allows studying the effects of 3D radiation transfer on clouds' dynamics and general properties like cloud cover, thickness and water density.

I think that the paper is suitable for publication in ACP, the analysis is complete and convincing, and the paper is clearly written. I believe that after a few clarifications the paper will be ready for submission, hence, I suggest a minor revision.

We want to thank the reviewer for the kind words about our manuscript and for taking the time to review our manuscript. We address the suggested clarifications below (in green text).

Major comments:
1. L.88: Can the authors explain why they chose a skin heat capacity of zero for the interactive surface and how realistic it is? To a none expert in the matter, it sounds like this could cause quick and unrealistic warming of the surface that can highly influence shallow convection.
   Little is known about what the most realistic skin heat capacity is for a grassland and what its impact on shallow convection is. Van Heerwaarden (2011) investigated the sensitivity of simulations of a convective boundary layer to the skin heat capacity and found that the heat capacity of the skin layer (which represents the vegetation) has minimal influence, because – even with a non-zero heat capacity -- the surface temperature responds very fast to the radiation, and the fastest fluctuations are anyhow mixed away by the turbulence. An instantaneous responding surface is used in previous LES work, e.g. by Lohou and Patton (Journal of the atmospheric science, 2014) and Gehrke et al. (GMD, 2021). Furthermore, previous work with MicroHH with the same set-up for the land surface model that we use shows that shallow convection is realistically modelled (van Stratum et al., JAMES, 2023). Hence, we chose to use the zero skin heat capacity. Please see also our answer at comment 7 of reviewer 3, where we discuss the land-surface model.
2. L.161-166: This part of the paper is unclear. Please explain if aerosols affect the dynamics of the simulations. In short, how are the microphysical processes handled? Are aerosol radiative effects and horizontal variability coupled to in the simulations?
   We thank the reviewer for pointing out that this part is unclear. In line 152 we mention briefly that we only include the direct effect of aerosols, by which we mean that the aerosols do not influence the microphysics. We will write this

more explicitly. The radiative effects and horizontal variability that comes from the horizontal variability in humidity is included in the coupled radiation calculations (and also in the uncoupled radiation calculations). We will explicitly mention this in line 166.

3. L.180-187: Please explain how cloud cover is defined. This is a tricky definition that can make comparisons between different datasets complicated (especially models to observations). Since the 1D and 3D don't show much difference, I suggest showing the sensitivity of 3D to different definitions or choices of thresholds.

We will add in these lines an explanation about how the cloud cover is defined and we will make it clear that we only make a rough comparison between the observations and the simulations, as these comparisons are tricky and we only aim to show that our simulations represent realistic conditions, we don't claim or aim to have an exact match. In addition, we shortly investigated the impact of the used definition in one of our 3D simulations as suggested by the reviewer and we will briefly mention the result hereof in the paper.

The figure below shows the cloud cover following the model definition (labelled ql_cover) for the example day from figure 5. One alternative option to define cloud cover in the simulations with 3D radiation is to look at the clouds under an angle. We can determine the cloud cover along the angle of the sun from the surface area that is shadowed with a shadow being an area with the global radiation less than 120 W m$^{-2}$. With this definition the cloud cover is higher when the solar zenith angle is large, and the cloud cover is lower when the solar zenith angle is small, but the differences are limited to +/- 0.04. With a stricter definition (global radiation less than 60 W m$^{-2}$ the cloud cover reduces, but the difference between the two thresholds is always smaller than 0.025. Therefore, for our rough comparison between the observations and our model a different definition leads to the same conclusion.

[Figure]

4. L.320: It is reasonable to assume that changes in cloud properties like cloud cover and optical thickness are most important for global surface radiation and scene albedo. The current version of Fig. 3 suggests that other processes

influence cloud cover more than the radiative transfer scheme (RT). This raises the question of how important 3D effects are for surface or total energy budget. Can the authors elaborate on this in the discussion and maybe even compare the bias caused by using 1D RT with other known biases and uncertainties in cloud or atmospheric modeling (like the choice of advection scheme, model resolution, microphysical scheme, etc.,)?

We agree with the reviewer that figure 3 shows that the cloud cover is not strongly influenced by the 3D radiation and that other known biases and uncertainties might have a larger influence on the simulated cloud cover. We will mention that the cloud cover might be more sensitive to other model choices such as the choice of advection scheme, model resolution, microphysical scheme when we discuss figure 3 (last paragraph of section 3). However, it is beyond the scope of our paper to investigate how sensitive the clouds are for these choices and subsequently how sensitive the energy budget is to these choices.

Minor comments:
1. L.147: I suggest using different names for the decomposed effects. Uncoupled is quite confusing and at the start can also be interpreted by the reader as 3D-1Drad3D. I would suggest something like Radiative-only. The cloud effect could be referred to as the 3D-coupling effect or dynamic effects.

   We understand the confusion about uncoupled as we use it both for uncoupled radiation computations and for the uncoupled effect. Therefore, we will rename the uncoupled effect to radiation effect. We feel like the name cloud effect is not causing confusion and can therefore be kept. In addition, we argue that it is important to have short and easy descriptions of these effects to keep the text readable, hence radiation-effect and cloud-effect work well.

2. L.198: Can the authors please explain why they chose to use the characteristic length scale and what is its physical meaning? Why wasn't a simpler measure of cloud size like mean size used?

   We will elaborate in the text on the meaning of the characteristic length scale. The advantage of the characteristic length scale is that it gives an indication of the size not just for the clouds, but for structures in general. Therefore, the larger characteristic length scales for vertical velocity in the boundary layer and specific humidity above the boundary layer show that the turbulent structures become larger, which inevitably means results in larger clouds. The disadvantage of the cloud size is that more assumptions are needed to determine it. One needs a tracking algorithm to identify all the clouds for which different options exist (see Heus and Seifert, GMD, 2013 and references therein). After tracking the clouds size needs be defined, which also can be done in different ways (see Mol et al., JGR Atmopheres, 2023).

3. L.209: I wonder what are the effects of these findings on the scene albedo (top of the atmosphere upwelling fluxes). If cloud cover is the same but the clouds are thicker, does it mean a larger cloud radiative effect?

   We agree with the reviewer that this is an interesting question and a nice addition to the paper. Therefore, we will add the top of domain upwelling fluxes in Figure 6

and add a short discussion of these results to the text. The similar cloud cover but thicker clouds indeed result in more radiation going upwards at the top of the domain (cloud effect). However, this is compensated by the radiation effect causing the net difference in upward radiation at the top of our domain to be close to zero.

4. L.224: Please explain what is the displacement distance. Does it change with the radiative transfer scheme? If LWP is higher then clouds might live longer and be more advected.
The displacement distance is the horizontal distance between a cloud and its shadow, derived from the domain-averaged cloud base height and the solar zenith angle (as described in section 2.2.2.). As we only used the term displacement distance a few times, we will write it out where we use it. This is only a relevant quantity for the simulations with 3D radiation, as with 1D radiation the shadow is always directly underneath the clouds, hence it is not horizontally displaced. The displacement distance is not related to how far the cloud moves during its lifetime.

5. L.241: Can the authors explain how the spread is quantified? Since the presentation is of only 3 cases, statistical measures are ambiguous, could be better to simply plot all three time series.
Since we have 3 repetitions of the simulations with coupled 1D and 3D radiation, we can make 9 combinations for the coupled effect and cloud effect. For the radiation effect there is indeed only three. We will explain this better around line 241 and we will adapt the caption of the figure.

6. L.149-255: It took me a minute to understand the discussion about the splitting methods. Might be clearer to mention the two methods by referring to Fig.1 or showing it mathematically (e.g., 3D-1Drad3D vs. 3Drad1D-1D).
We will add the mathematical description here and refer to figure 1.

7. L.322-326: I think that the authors can show the role of 3D radiative transfer on Earth's energy budget with not a lot of extra effort. What are the changes in atmospheric heating and top of the atmosphere fluxes? Does decreased diffused radiation on the surface means increased heating rates in the atmospheric or higher scene albedo at the top-of-the-atmosphere? Might be worth to have even a short discussion on this as well.
We thank the reviewer for this nice suggestion. We will add the top of domain upwelling fluxes in Figure 6 and add a short discussion of these results to the text. To link the results at the surface to what happens at the top of the domain, one should not just consider the diffuse radiation, but also the direct radiation that is scattered by the surface and goes back up (as diffuse radiation), hence the top of domain radiation is most closely related to the global radiation.

8. L.336: Is there a reason to assume that the findings of this paper will be different away from the mid-latitudes? Dror et al., (IEEE, 2020) showed that a dominant subset of such clouds doesn't have a strong latitudinal dependence.
We thank the reviewer for bringing this paper to our attention. Our main reason to expect a different effect at different latitude is (as mentioned) that other latitudes have other solar zenith angles, which will impact the 3D effects. However, we agree with the reviewer that the results of Dror et al form a reason to

expect similar results in other regions, hence we will mention both possibilities in the paper.

Technical comments:
1. L.17: In cloud and weather modeling communities Cloud resolving models are usually referred to course resolution models on a scale of 1 km.
For these models it also holds that the cloud and its shadow will not be located in the same grid box, hence 3D effects are relevant. We will rephrase this such that it refers to both cloud resolving models and large eddy simulations.
2. L.90: Worth mentioning that RRTM is 1D, and explain, even in short, the ray tracing concept and the novelty of the GPU usage (in Veerman et al., 2022 line 90).
We will mention that RRTMGP is 1D and we will explain briefly the novelty of the ray tracer of Veerman et al., 2022.
3. L.160: This is not very clear, does aerosol vertical profile change with time in simulations?
Yes, it does, we will formulate this clearer.
4. 5: adding y-axis labels as in Fig.6 would make the figure clearer.
We left out the labels in figure 5 on purpose. Figure 5b shows 3D-3Drad1D (as in figure 6), but also 1Drad3D-1D. Similarly, figure 5c shows 3Drad1D-1D as in figure 6, but also 3D-1Drad3D. Putting all of this in the y-axis labels makes it to our opinion only less clear.
5. 6 captions: Which dataset is presented, worth mentioning it's for all 12 days.
We will mention this as suggested.

Additional references
- van Heerwaarden, C. C. (2011). Surface evaporation and water vapor transport in the convective boundary layer. Wageningen University and Research. (https://edepot.wur.nl/169077)
- Gehrke, K. F., Sühring, M., & Maronga, B. (2021). Modeling of land–surface interactions in the PALM model system 6.0: land surface model description, first evaluation, and sensitivity to model parameters. Geoscientific Model Development, 14(8), 5307-5329.
- Heus, T., & Seifert, A. (2013). Automated tracking of shallow cumulus clouds in large domain, long duration large eddy simulations. Geoscientific Model Development, 6(4), 1261-1273.
- Mol, W. B., van Stratum, B. J., Knap, W. H., & van Heerwaarden, C. C. (2023). Reconciling observations of solar irradiance variability with cloud size distributions. Journal of Geophysical Research: Atmospheres, 128(5), e2022JD037894.

---

## Author Comment (AC3)

**Review of "The impact of coupled 3D radiative transfer on surface radiation and cumulus clouds over land" by Tijhuis et al.**

General comments

This paper presents large eddy simulations of continental cumulus clouds with a coupled shortwave 3D radiation scheme. Results are compared to simulations with a standard 1D radiation scheme to assess the impact of shortwave 3D radiation effects on the clouds and surface radiation. Simulations with 3D radiation are found to have larger and deeper clouds, which reflect more shortwave radiation and therefore reduce the surface shortwave radiation. This acts in the opposite direction to non-coupled 3D radiation effects that tend to enhance the surface shortwave radiation. Overall, there is an almost exact compensation such that the mean surface shortwave radiation in simulations with 1D and 3D radiation is very similar. This finding is based on multiple simulated days.

The paper is clear and well written. I learned a lot from reading this paper. I would like to congratulate the authors on their significant new findings. The conclusions are mostly convincing, and not necessarily expected. I think this paper will spur further studies and will become a well cited reference in the coming years. I have a few suggestions to further strengthen the study, as outlined in my comments below. After addressing these comments, I believe the study will be appropriate for publication in ACP.

We want to thank the reviewer for the kind words about our manuscript and for taking the time to review our manuscript. We address the given comments below (in green text).

Specific comments
1. Title: I suggest changing to "The impact of coupled shortwave 3D radiative transfer on surface radiation and cumulus clouds over land" because the authors did not consider coupled longwave. This is already mentioned in the manuscript, but it would add clarity to put this extra word in the title.
   We will add this to the title as suggested.
2. Introduction: Several references are made to "cloud resolving models". I think the authors are only referring to large eddy simulations (LES). Often, the term "cloud resolving model" is used to refer to a model that is run without a convection scheme. These models can still have horizontal resolution of a few km, but I don't think that's what the authors intend. I recommend saying LES from the beginning, or even better explicitly state the range of horizontal resolutions that are used in the cited studies.
   For these models it also holds that the cloud and its shadow will not be located in the same grid box, hence 3D effects are relevant. We will rephrase this such that it refers to both cloud resolving models and large eddy simulations.
3. L47-49: It is interesting that the study from Jakub and Mayer claimed that cloud organization (cloud streets) occur with coupled 3D radiation, but the studies by Veerman et al. and the present study do not seem to identify or even mention cloud organization. I think it is worth mentioning this difference in the

introduction and/or conclusions as a potential discrepancy that exists in the literature.

We agree with the reviewer that it would be interesting to identify the cloud organization to compare with the results of Jakub and Mayer. Unfortunately, determining if and how the clouds organize in our cases is more complex than in the cases of Jakub and Mayer. Jakub and Mayer use fixed solar azimuth angles and wind directions, which limits the formation of cloud streets to the north-south and east-west direction. This allows for an intuitive quantification of the cloud organization. In our cases (and also in the ones in both studies of Veerman et al) the solar azimuth angle changes continuously following the daily cycle of the sun and the wind direction is never perfectly in the north-south or east-west direction. Therefore, it is not trivial to describe the cloud organization, and one would need to carefully examine different methods to describe cloud organization, which is beyond the scope of the present study. However, we will elaborate the indicated lines to highlight the differences between the setups better, as the different setups explain why cloud organization as it is described in Jakub and Mayer is not considered in the present study. In addition, we will shortly describe in section 4.1. why we do not investigate the cloud organization as they do in Jakub and Meyer.

4. L75-76: It would be helpful to expand on what is meant by "the days where the simulated cloud cover visually matches the observed cloud cover". What is the criteria for a match? For example, does the match consider only cloud cover or also cloud size, shape, organization, etc?

We will expend on the meaning of 'visually match'. Our criterium for a match is that there is no systematic under or overestimating of the cloud cover by tens of percents, which can happen e.g. when the clouds are forced by a large-scale system that is not captured by the LES. It only considers cloud cover, as we only have observations of cloud cover.

5. Section 2.1: I recommend listing the exact dates that were used for the clear-sky (13 days), cloudy (20 days), and subset cloudy (12 days). Or put them in a table. This is needed for reproducibility reasons.

We will add a list of dates to the complete model setup in the zenodo repository.

6. L82: I do not see any supplementary materials uploaded. Can the details of the LES model be referred to previous literature?

Our apologies for this mistake, this should refer to the zenodo repository that is mentioned in the code and data availability statement. We will put the correct reference in the text.

7. L88: A couple of sentences explaining what the land surface model is actually doing would help here. How realistic is the assumption of an instantaneous surface response? I am concerned that this assumption might make the coupling with the clouds too strong. If, in reality, there is some delay of the surface response, then the clouds could evolve or be advected before they "feel" the surface immediately below them. I would have thought that the land surface model needs to account for the processes that determine the response time of the surface such as heat transfer between vegetation and soil layers, and stomatal opening/closing.

We will add a couple sentences to the manuscript that describe the land surface model. The land surface model includes heat transfer between the skin layer and the soil layers below. Because of this coupling, part of the surface response is actually delayed. This works in the following way: a peak in radiation causes an immediate heating of the skin layer, which causes a large soil heat flux. The soil layer that receives this heat flux warms up slowly and can provide an upward flux back to the skin layer at a later moment. Hence, in a way part of the surface response is delayed. We agree with the reviewer that for the most accurate response of the surface, the stomatal opening/closing should be considered, but this is not included in our land surface model. This would require the addition of a plant model, as is e.g. used by Sikma and Vila (GRL, 2019). Please see also our answer at the first major comment of reviewer 2, where we discuss the skin heat capacity.

8. L90: Can some justification be provided that the radiation calls once every minute are sufficient? The appendix shows that the wind speed often exceeds 5 m/s at 10 m altitude. The wind speed at cloud altitude is probably even larger. Taking 5 m/s as a typical wind speed, in the 1 minute between radiation calls a cloud would be advected 300 m. Figure 4d shows that 300 m is comparable to the length scale of the simulations. This means that an individual cloud will move a distance that is comparable to the size of its shadow before the position of the shadow is updated in the simulation, which seems quite "jumpy". Given that the study depends critically upon the land-atmosphere coupling imposed by the evolving pattern of cloud shadows, I think the reader needs some more convincing on this decision.

To test the sensitivity to the radiation timestep we performed one simulation with 1D radiation and one with 3D radiation for the chosen example day in figure 5 with the radiation called every 15 seconds. The differences between these two runs are given by the dashed lines in the plot below. Apart from the dashed lines, this plot is identical to figure 5 in the manuscript.

[Figure]

Most of the time, the difference between the simulations with 15 second radiation is within the range that we found for the simulations with 1 minute radiation. Around 10 o'clock the differences with 15 second radiation are slightly outside of the range that we found before. However, these minor changes in the

radiation differences will not change our conclusions. We will shortly mention this in the manuscript.

9. L108-112: This hypothesis would benefit from some discussion of the time/length scales of the boundary layer mixing. The surface is not instantaneously connected to the cloud immediately above it. It takes time for perturbations in surface fluxes to be transported up to cloud base. And during this transport there must also be some mixing that occurs, such that the variability at cloud base is not as sharp as the surface. For the clouds to be influenced, I think it has to be the case that:

   1. the timescale for mixing to cloud base is shorter than the timescales of individual cloud evolution and movement, and

   2. the surface discontinuities are not simply mixed away during transport to the cloud layer.

   For example, one recent study that considered how these types of clouds change during a solar eclipse suggested that the fastest timescale for surface air parcels to be transported to cloud altitude is around 15 minutes (https://doi.org/10.1038/s43247-024-01213-0). It is not clear that this is shorter than the timescales of cloud movement and growth/decay, which leads me to question this hypothesis.

   It is hard to give a general discussion about the time and length scales involved because of the large range of scales involved. As figure 4d shows, the length scales involved differ strongly between the individual dates and times. Heus and Seifert (gmd, 2013) showed that the lifetime of shallow cumulus can range from less than 1 minute up to 120 minutes. The recent study about the solar eclipse also shows that the updraft speed continuously changes during the day. Altogether this indicates that to test our hypothesis further than what we did now, one should look at the individual days/times/clouds. Only this would allow to determine if and when the updraft speed and location match the cloud movement and development. To do so is tricky with the current setup because of the potential influence that the clouds have on each other. Therefore, a more idealistic setup might yield further insights (as we suggest in line 333), but this is beyond the scope of the present work.

   We agree with the reviewer that some of the surface variability is mixed away during the upward transport, however definitely not all the variability is mixed away. If all the variability would be mixed away before the cloud layer is reached, there would be no 3D effect because of the surface heterogeneities. Veerman et al 2022 showed that when the surface radiation is homogenized, the simulations with 1D and 3D radiation give nearly identical clouds. Hence, the surface discontinuities are key to get a difference between simulations with 1D and 3D radiation.

10. L158: Give -> Given

    We will change this as suggested.

11. L161-162: At large RH, above 90% or so, even a small change in RH can result in a large change in optical properties due to the non-linearity in aerosol extinction as a function of RH. Are the optical properties defined per 10% RH even at high RH? If so, this likely introduces an important source of error for aerosol optical properties in the vicinity of clouds. These errors will not be evident in the clear

sky cases that are used for validation, but they will be present on cloudy days because RH will approach 100% toward cloud edges.

Our apologies for the incomplete description. The optical properties are defined per 10% RH between 0 and 80% RH, after that, the properties are described per 5% RH. We will correct this in the manuscript.

The intervals indeed introduce an error at high rh because of the non-linearity in aerosol optical properties. However, the error that remains with the 5% classes is likely small compared to the differences between observed and simulated radiation caused by differences between observed and simulated clouds. As our main aim with the aerosol implementation is to remove the systematic bias that we had before, we argue that the 5% classes are sufficient.

12. L172: It should be noted that the definition of cloud cover in observations and models is slightly different here. A scanning or wide-field view instrument will detect and include cloud sides as part of the overall cloud cover. In contrast, cloud cover in the LES model is defined only from a zenith view perspective. This will generally lead to an overestimate of cloud cover in observations relative to LES, unless an instrument simulator is used within the LES to ensure sampling consistency (which I don't think is done here). This fact also has implications for one of the main results of the study, that the clouds are deeper in 3D coupled radiation but the cloud cover is the same. This may be true with the model definition of cloud cover. But from an observational perspective, the cloud cover could actually still increase because the deepening of clouds results in more of the sky becoming obscured at oblique views of the instrument. This is worth commenting on in the conclusions.

The reviewer is correct that we don't have an instrument simulator in our LES and the definition of cloud cover is different from an observational perspective compared to the model. We are aware of this difference and therefore we don't aim for a perfect match between observations and simulation. We will mention this difference when we discuss figure 3 (last paragraph of section 3). We will also explicitly mention here that we use the model definition of the cloud cover in the remainder of the paper.

13. L211-223: Can a statistical significance test be done to determine whether the correlations presented are significant? That would make the presented correlations more convincing.

We thank the reviewer for this suggestion, and we agree that this makes the correlations more convincing. We tested the significance and found the following results:
- for shadow displacement: r = 0.266, p = 3.32e-09
- for wind-sun-angle: r = 0.554, p = 4.32e-40
- for wind speed: r = 0.330, p = 1.153e-13

Hence the correlations are significant, which we will mention in the manuscript.

14. L211-231: Similar to my comment above about the hypothesis for cloud changes, I think this discussion would benefit from considering the time and space scales involved.

Please see our consideration of the time and length scales at the comment above.

If the hypothesis holds, the correlation should be highest for a combination of the factors explored: when the wind direction is aligned with the sun angle AND the shadow displacement divided by the wind speed is similar to the cloud base height divided by the updraft speed. Could the authors look at this explicitly and see if they find a connection?

We defined the the time scale mismatch as the shadow displacement divided by the wind speed minus the cloud base height divided by the updraft speed. If we understand the comment correctly, the reviewer suggests that there should be a high correlation between the difference in liquid water path and the combination of the wind sun angle and this time scale mismatch. We tested this and found a correlation coefficient of 0.56, which is practically the same correlation as we found with the wind sun angle alone.

We agree with the reviewer that if the wind sun angle is small and the time scale mismatch is small, the clouds in the simulations with 3D radiation feel their own shadows most, hence the difference in liquid water path might be small. However, when a cloud moves away from its own shadow or perpendicular to its own shadow, a matching time scale does not necessarily result in a stronger or weaker cloud development. This will depend on the location of the shadows of other clouds and potentially on the location of the strongest cloud enhancements. Therefore, adding the time-scale mismatch does not add to the correlation that is found with only the wind sun angle.

It would be interesting to look at only cases where the wind direct is aligned with the sun angle to see if indeed a smaller time scale mismatch results in a smaller 3D effect. Unfortunately, if we select only the times where the wind direction is aligned with the sun angle (difference in angle < 25 degrees), only 1/8 of our dataset remains and the variation in time scale mismatch is very limited. A more idealized setup (as we suggest at the end of our manuscript) where e.g. the wind-direction and sun angle are fixed might therefore yield more insight.

I also wonder if it is possible that this combination of factors could lead to a suppression of cloud development in 3D in the case that clouds are systematically moving toward their shadows in 3D. Do the authors see any evidence of this? If not, does this provide evidence to reject the proposed hypothesis?

We agree that the combination of factors could lead to the suppression of cloud development in 3D if the clouds are moving towards their own shadows. The opposite should be the case when the clouds move away from their own shadows. Both must be the case to have a good correlation between the difference in liquid water path and the wind-sun angle. A small wind-sun angle (the cloud moving towards its own shadow) should correspond to a small difference in liquid water path and opposingly a large wind-sun angle (the cloud moving away from its own shadow) should correspond to a large difference in liquid water path. The significantly positive correlation between wind-sun angle and differences in liquid water path provides some evidence that this is the case. However, it is important to note that this theory only takes into account the impact of the clouds own shadow, whereas clouds can in our setup also be influences by shadows of other clouds, hence we cannot expect a perfect correlation.

15. Fig. 6: Are the box plots showing the mean across the entire day or at a specific time during the day? I might have missed it but I don't see this mentioned anywhere.

   It shows all times with clouds. We will mention this in the caption of the figure.

16. Section 4.2.1 and Fig. 5b: The global uncoupled effect is always positive, meaning that the diffuse effect dominates throughout the day. The Gristey et al paper that is already cited showed that the global effect can be negative at the end of the day on some days, because the direct effect can dominate at oblique sun angles. I am curious, do the authors see this on any of their simulated days, or is the global effect positive for all times of day and all cases? This would be an interesting similarity or difference to note in the paper.

   We thank the reviewer for bringing up this similarity between our work and the work of Gristey et al. As can be seen from the boxplot in figure 6b, there are some timesteps with a slightly negative global uncoupled effect (or radiation effect as it will be named after the revisions). These moments are indeed at the end of some of the days. We will shortly mention this in relation with the Gristey et al paper.

17. Schematic figure of key result: I think this paper would really benefit from a schematic figure that captures the main result in the abstract ie. the almost exact compensation between uncoupled and cloud 3D effects. I encourage the authors to consider creating a schematic that represents this result in an intuitive and concise way. Figure 1 achieves this for the methodology. I am thinking of something similar to Figure 1, but for the results. This type of figure can help to engage a broader audience and increase the impact of the study.

   We noticed that most papers in ACP support their abstract with a complete figure from the paper. We agree with the reviewer that something more concise is helpful for a broader audience, hence we opted for a simplified version of figure 6 that only shows the main results i.e. the almost exact compensation between uncoupled and cloud 3D effects.

[Figure]

Additional references
- Sikma, M., & Vilà-Guerau de Arellano, J. (2019). Substantial reductions in cloud cover and moisture transport by dynamic plant responses. Geophysical Research Letters, 46(3), 1870-1878.
- Heus, T., & Seifert, A. (2013). Automated tracking of shallow cumulus clouds in large domain, long duration large eddy simulations. Geoscientific Model Development, 6(4), 1261-1273.

---

## Author Response (AR2)

We thank the reviewers for their time and effort to reviewer our manuscript.

We addressed the final concern of reviewer 2 by adding a short description of the representation of vegetation in our model to the second paragraph of section 2.2.1. Our land-surface model takes into account changes in the canopy resistance as a function of incoming shortwave radiation, soil moisture, and vapor pressure deficit. Hence, our vegetation does respond to the changes in radiation that our paper focusses on. Although more advanced vegetation models exist, we argue that our setup is suitable for our purpose, as van Stratum et al. (JAMES, 2023) showed that the modeled surface fluxes agree with the observations.